# The Association between mHealth App Use and Healthcare Satisfaction among Clients at Outpatient Clinics: A Cross-Sectional Study in Inner Mongolia, China

**DOI:** 10.3390/ijerph19116916

**Published:** 2022-06-05

**Authors:** Li Cao, Virasakdi Chongsuvivatwong, Edward B. McNeil

**Affiliations:** 1Information Technology Department, Inner Mongolia Medical University, Hohhot 010110, China; licao@immu.edu.cn; 2Department of Epidemiology, Faculty of Medicine, Prince of Songkla University, Hat Yai 90110, Thailand; edward.m@psu.ac.th

**Keywords:** client, satisfaction, mHealth app use, structural equation modeling

## Abstract

Mobile health (mHealth) applications (apps) have been developed in hospital settings to allocate and manage medical care services, which is one of the national strategies to improve health care in China. Little is known about the comprehensive effects of hospital-based mHealth app use on client satisfaction. The aim of this study was to determine the relationship between the full range of mHealth app use and satisfaction domains among clients attending outpatient clinics. A cross-sectional survey was conducted from January to February 2021 in twelve tertiary hospitals in Inner Mongolia. After the construction of the mHealth app use, structural equation modeling was used for data analysis. Of 1889 participants, the standardized coefficients β on environment/convenience, health information, and medical service fees were 0.11 (*p* < 0.001), 0.06 (*p* = 0.039), and 0.08 (*p* = 0.004), respectively. However, app use was not significantly associated with satisfaction of doctor–patient communication (β = 0.05, *p* = 0.069), short-term outcomes (β = 0.05, *p* = 0.054), and general satisfaction (β = 0.02, *p* = 0.429). Clients of the study hospitals were satisfied with the services, but their satisfaction was not much associated with mHealth use. The limitation of the mHealth system should be improved to enhance communication and engagement among clients, doctors, and healthcare givers, as well as to pay more attention to health outcomes and satisfaction of clients.

## 1. Introduction

Over the last decade, mobile technologies and their applications have been increasingly used in the healthcare field, specifically in the field of mobile health (mHealth) [1,2]. During the COVID-19 pandemic, mHealth has been a critical digital tool for maintaining social distance and facilitating testing, tracing, and isolation [3,4,5]. mHealth apps have been developed in hospital settings to allocate and manage medical care services, as well as to improve patient satisfaction by utilizing all available facilities and medical care data [6]. The most frequently used functions are those for providing and managing patient care in larger hospitals, as well as informative hospital applications that assist with routines such as admissions, check-in, billing, doctor–patient communication, training, education, discharge, and check-out [6,7]. Certain hospitals have begun using online medications and consultations to deliver healthcare [8,9]. Around 83% of Chinese tertiary hospitals use mHealth apps to schedule appointments and provide other services [6]. Of them, appointment rates for seeing a doctor are required to reach more than 50%, according to a national policy [10]. Scaling up mHealth is one of the Chinese government’s strategies for providing affordable, accessible, and appropriate health care to the general public [11].

Patient satisfaction has emerged as a critical indicator of healthcare quality [12,13] and healthcare policy [14,15], which are associated with process quality, readmission and mortality rates of surgical care in US hospitals [16]. Hospital patient satisfaction is the result of an integrative process that includes not only concerned high-quality doctors but also enhanced convenience features such as an easy-to-use reservation system and comfortable waiting areas [17]. Authors of a systematic review concluded that the construct of healthcare satisfaction should be measured using a multidimensional approach, with the physical environment, patient–doctor communication, and hospital management processes serving as the primary domains of many instruments [18]. Overall satisfaction in hospitals is largely determined by outpatient satisfaction [19,20], which could be improved by effective patient–doctor communication [20,21], decreased medical costs, convenient medical treatment process and hospital environment, and shortened waiting time for medical services [22,23,24,25]. Whether mHealth can improve satisfaction levels is not well understood.

Some practices of web-based appointment systems have reported positive results, including reductions in no-show rates and waiting times [26] and increased patient satisfaction [10,27]. As an update to web apps, mHealth apps can be more patient-centered [28] and offer more benefits for improving access and communication between patients and doctors in real time [29,30,31], which is expected to increase patient satisfaction across multiple dimensions, and finally facilitate health outcomes [32,33]. mHealth has altered patient procedures in the hospital, starting with a convenient paperless registration service and more effective workflow [34,35], which has significantly reduced outpatient waiting times and increased patient satisfaction [31,36,37]. A smartphone is an under-utilized tool that can enhance patient–physician communication, increase satisfaction, and improve the quality of care [38]. A mHealth-based virtual clinic has the potential to support patients during their consultation with health professionals [36].

Previous studies have mainly reported the effects of mHealth app use on appointment scheduling and registration process via one-step regressions [31,32,36,37]. Little is known about the comprehensive effects of the full range of hospital-based mHealth app use on healthcare satisfaction in outpatient departments, such as doctor–patient communication, health information, medical service fees, and short-term outcome. Structural equation modeling (SEM) is a powerful analytic tool to examine complex causal models among multiple variables simultaneously and use latent factors to reduce measurement error, which is superior to one-step regressions [37,39]. Our multidimensional analysis aims to examine the relationships between the full range of mHealth use and each domain of outpatient department satisfaction using SEM. The results of this study will allow mHealth providers to improve their apps in accordance with the patient satisfaction standard.

## 2. Materials and Methods

### 2.1. Study Design and Setting

A cross-sectional survey was carried out in Inner Mongolia of China from January to February 2021. Inner Mongolia is located in the northern part of China, where traditional Mongolian medicine, traditional Chinese medicine, and western medicine are well distributed and accepted by local citizens [40]. Similar services in mHealth apps are provided among the studied hospitals, including electronic health code (eHealth code), appointment, consultation, payment, record checking, and healthcare rating.

### 2.2. Participants

Clients at outpatient departments (OPD) aged 18 years or above, able to speak Mandarin, and undergoing non-emergency treatment were eligible for the study.

### 2.3. Procedure

A team of resident physicians from Inner Mongolia Medical University was trained for data collection. Clients were consecutively approached at departure areas or at the main pharmacy, explained about the study, and asked whether they would like to participate in the survey. Face-to-face interviews based on the questionnaire were completed in around 15 min. The research study was approved by the Office of Human Research Ethics Committee, Faculty of Medicine, Prince of Songkhla University (REC.63-306-18-1), and Inner Mongolia Medical University (REC.YKD202201096).

### 2.4. Variables

mHealth app use was considered a latent variable consisting of the following observed yes/no items: having a mHealth app, having an e-health code, history of making an appointment with doctors online, using e-payment, health record checking, consultation, and healthcare rating.

Client satisfaction was assessed using the Chinese Outpatient Experience Questionnaire, which included 28 items measuring 6 dimensions, namely physical environment and convenience, doctor–patient communication, medical service fees, health information, short-time outcome, and general satisfaction [41]. The responses to each item were rated on a 5-point Likert scale (1 representing the worst satisfaction and 5 representing the best satisfaction). For descriptive analysis, the average of all the items within each dimension was used. The total satisfaction score was the average score of all 28 items.

Covariates included social-demographic variables such as age (in years), gender(male or female), area of residence (urban or rural), education level (primary or less, secondary, tertiary), monthly household income (in yuan), which was categorized into 6 income groups (0–2000, 2001–4000, 4001–6000, 6001–8000, 8001–9999, and 10,000 or more), and an indicator variable defined as 1 for first-time visitors to the hospital, otherwise 0.

### 2.5. Statistical Analysis

Data entry and validation were performed using EpiData 3.1 [42], and data analysis was conducted using R version 4.0.1 [43]. Frequencies and percentages were used to describe categorical variables, whereas means and standard deviations were used to describe continuous variables.

Exploratory factor analysis (EFA) and confirmatory factor analysis (CFA) were used to analyze mHealth use and the correlations among the dimensions of client satisfaction, respectively. The association between sociodemographic variables, mHealth app use, and satisfaction was examined using a multiple indicators, multiple causes model (MIMIC) with structural equation modeling (SEM) [44]. The “psych” and “Lavaan” packages in R were used for EFA [45] and CFA/SEM [46], respectively. The sample size was calculated with a 10% anticipated effect size [31], a desired statistical power level of 80%, and a 95% confidence interval [47]. In total, 7 latent variables, 35 observed variables, and a 15% non-response rate were assumed. The minimum required sample size was 2128. Of the 2182 eligible clients, 1889 were included, as 293 were excluded for not owning a smartphone and incomplete survey responses.

## 3. Results

### 3.1. Demographic and Economic Factors

As shown in Table 1, the mean (SD) age of all participants was 42.3 (14.4) years; nearly half were female, the majority lived in an urban area, and nearly half had a tertiary education level. The median household monthly income was 4000–6000 yuan, which was considered the middle class in China [48].

### 3.2. mHealth App Use

Overall, 65.5% of clients owned an mHealth app, 71.0% had an eHealth code, and 55.4% could make an appointment to see a doctor using the app. Nearly half used e-payment for healthcare, 35.2% reviewed their health record on a mHealth app, around 12% consulted with a doctor, 16% rated healthcare services online, and 17.4% never used any mHealth services.

### 3.3. EFA Model of mHealth App Use

The Kaiser–Meyer–Olkin (KMO) measure was 0.87, which was above the recommended threshold of 0.6, and Bartlett’s test with (χ^2^ (21) = 1408.78, *p* < 0.001) was significant, indicating that the data had sampling adequacy for EFA [49]. We used Velicer’s minimum average partial (MAP) criterion to achieve a minimum of 0.05 with 1 factor [50]. The appropriate extraction method was principal axis factoring (PAF) [51], and the appropriate oblique rotation method was “oblimin” [49] since the data were not normally distributed. Details of the EFA are shown in Table 2. The factor loadings ranged from 0.70 to 0.97. Cronbach’s alpha was 0.83. The mHealth app use factor explained 71% of the total variance of the domain. These results indicated that the construct was adequate.

### 3.4. CFA Model of Client Satisfaction

The measurement model of client satisfaction was adequately explained by items with high factor loadings, as shown in Table 3. The correlation matrix with average variance extracted (AVE) is shown in Table 4. Overall client satisfaction had an average score of 4.07 out of 5, and there was little variation across the six dimensions. The model fitted the data well with relative Chi-square = 2.854 (χ^2^/df), comparative fit index (CFI) = 0.958, and Tucker-Lewis index (TLI) = 0.953, root mean square error of approximation (RMSEA) = 0.031 (90% CI: 0.029, 0.031), and standardized root mean squared residual (SRMR) = 0.034. Cronbach’s alpha reliability coefficient was greater than 0.7 and average variance extracted (AVE) was greater than 0.5.

Since most of variables were categorical, a weighted least square mean and variance adjusted (WLSMV) estimator was used in the MIMIC model to investigate the associations among mHealth app use and client satisfaction [52]. All indices suggested that the model fitted the data well [53], with relative Chi-square = 2.48 (χ^2^/df), RMSEA = 0.028 (90% CI: 0.026, 0.030), SRMR = 0.031, CFI = 0.961, and TLI = 0.955.

Figure 1 shows the results of the regression of mHealth app use with each dimension of client satisfaction. The standardized coefficients (β) and 95% confidence intervals (CI) are shown in Table 5. The standardized coefficients (95% CI) on environment/convenience, health information, and medical service fees were 0.11 (*p* < 0.001), 0.06 (*p* = 0.039), and 0.08 (*p* = 0.004), respectively. In summary, mHealth app use did not significantly influence the satisfaction with doctor–patient communication, short-term outcome, or general satisfaction.

## 4. Discussion

Based on the government’s target for mHealth use of 50–100%, our study hospitals achieved the targets of having mHealth, having e-health code and appointment, but not on payment, record-checking, consultation, or online healthcare rating. Overall satisfaction with healthcare services was good, with average scores of around 4.07 out of 5 among clients visiting an OPD, and there was little variation across the six dimensions. Three dimensions of satisfaction, namely environment/convenience, health information, and medical service fees, were associated with mHealth app use. The coefficients between mHealth app use and these satisfaction domains were weak at around 10%; therefore, the effect of mHealth apps on client satisfaction was minimal.

In 2015, a national survey from the US reported that 58.2% of mobile phone users had downloaded a health-related mobile app [54]. The adoption rate of mobile services for outpatients was only 31.5% from a 2019 Chinese study [55]. Another study from Germany in 2017 found that 33.5% of outpatients admitted to using their mobile devices to manage their health-related data [56]. Our study had a slightly higher prevalence of having mHealth and mHealth use than those studies. All studies, however, suggested a fairly large proportion of respondents had not used mHealth apps.

Two Chinese studies reported an outpatient satisfaction score of 4.42 (out of 5) in 2015 [57] and 3.75 in 2021 [20], respectively. The differences in scores between these two studies were due to the different settings. Our average score of 4.07 was in the middle of these studies.

Similar to other studies, mHealth was effective in reducing patient waiting times and increasing patient satisfaction in tertiary hospitals [31,37,38]. Another study found that waiting times for consultations and prescription filling were reduced, resulting in increased satisfaction with outpatient pharmacy services [58]. Our study validated the marginal effect of mHealth app use on waiting time within environment/convenience. We did not find that the use of mobile health apps improved all six dimensions as found in a previous study [31]. There was also no significant relationship between short-term outcome, general satisfaction, and patient–physician communication. The development and usage of mHealth apps may still be at the initial stage, and therefore, it may be too early to gauge its real effect.

On patient–physician communication aspects, the overall satisfaction score was 4.13; however, there was little (β = 0.05) association with mHealth use. This may be because the use rates of consultation and rating in mHealth were low (12% and 16%, respectively). Effective communication plays an important role in patient-centered care and improves patient satisfaction [57,59]. The next development of mHealth should therefore focus on communication and consultations among patients and physicians on healthcare providers’ performance to further improve healthcare use, patient health outcomes, and satisfaction [60].

Some study limitations should be mentioned. First, the cross-sectional design limits the assessment of causality. Second, our findings were based on an ongoing stage of mHealth development; thus, further studies to follow up mHealth use, especially for doctor–patient communication, are needed.

## 5. Conclusions

Data showed that clients of the study hospitals were satisfied with the services, but their satisfaction was not much associated with mHealth use. On the one hand, the healthcare system of hospitals should continue to maintain such high satisfaction levels. On the other hand, the limitation of the mHealth system should be improved to enhance communication and engagement among clients, doctors, and healthcare givers, as well as to pay more attention to the health outcomes and satisfaction of clients.

## Figures and Tables

**Figure 1 ijerph-19-06916-f001:**
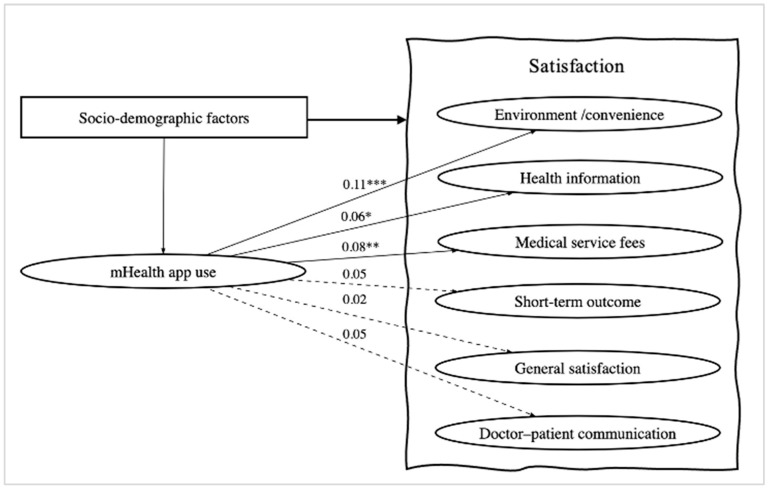
Structural equation model depicting the association of mHealth app use with client satisfaction. Note: Solid lines for significant relationships, dotted lines for non-significant ones; numbers on lines are standardized coefficients from mHealth app use to each domain of satisfaction. *** *p* < 0.001, ** *p* < 0.01, and * *p* < 0.05.

**Table 1 ijerph-19-06916-t001:** Sociodemographic characteristics of participating clients.

Variable	*n* (%)
Age (years), mean (SD)	42.3(14.4)
Gender	
Male	908 (48.1)
Female	981 (51.9)
Area of location	
Rural	410 (21.7)
Urban	1479 (78.3)
Education	
Primary or less	195 (10.3)
Secondary	672 (35.6)
Tertiary	1022 (54.2)
Monthly household income (RMB)	
0–2000	153 (8.1)
2001–4000	372 (19.7)
4001–6000	420 (22.2)
6001–8000	309 (16.4)
8001–9999	304 (16.1)
≥10,000	331 (17.5)
First visit	
Yes	496 (26.3)
No	1393 (73.7)
Having an mHealth App	
Yes	1239 (65.5)
No	650 (34.5)
eHealth code	
Yes	1342 (71.0)
No	547 (29.0)
Appointment online	
Yes	1046 (55.4)
No	843 (44.6)
Consultation online	
Yes	225 (11.9)
No	1664 (88.1)
E-payment	
Yes	937 (49.6)
No	952 (50.4)
Record checking	
Yes	664 (35.2)
No	1225 (64.8)
Rating on healthcare	
Yes	308 (16.3)
No	1581 (83.7)
Never use any mHealth	
Yes	328 (17.4)
No	1561 (82.6)

RMB: Chinese Renminbi.

**Table 2 ijerph-19-06916-t002:** Measurements of mHealth app use items and their reliability by exploratory factor analysis.

Dimension	Item	Loading	Communality
mHealth app use	Having mHealth apps	0.86	0.74
	Having e-health code	0.70	0.49
	Appointment of doctors	0.90	0.82
	Using e-payment	0.97	0.95
	Health record checking	0.93	0.86
	Consultation online	0.77	0.59
	Rating on healthcare	0.73	0.53

**Table 3 ijerph-19-06916-t003:** Items of client satisfaction and factor loadings by confirmatory factor analysis.

Dimension	Item	Loading
Environment/convenience	1. Waiting time was short	0.702
	2. Registration procedure was easy	0.750
	3. Dispensary/payment was convenient	0.732
	4. Visiting instructions and signs were clear	0.814
	5. Physical environment of OPD was clean	0.771
	6. OPD was quiet	0.689
Doctor-patient communication	7. Doctors explained things clearly and understandably	0.833
	8. Doctors listened to you carefully	0.839
	9. Enough time to communicate with doctors	0.801
	10. Doctors treated you with courtesy and respect	0.821
	11. Doctors cared about your anxieties or fears	0.825
	12. You were involved in decision making about treatment	0.833
	13. Your opinions/thoughts were respected by doctors	0.856
	14. Doctors protected your personal privacy	0.783
Health information	15. You received explanations concerning your illness	0.859
	16. You received information about the signals of dangerous conditions related to your illness when you were at home	0.838
	17. You received health knowledge related to your illness	0.814
	18. You received explanation about the following examination or treatment	0.861
	19. You received explanation about the results of examination/test	0.864
	20. Doctors explained the drug effects in a way you could understand	0.815
	21. You received medication precautions (directions and dosage, side effects, contraindications, etc.)	0.817
Medical service fees	22. Charges of this visit were reasonable	0.820
	23. Charges of this visit were transparent	0.888
	24. You could not afford the expenses of this visit	0.839
Short-term outcome	25. This visit could help you reduce or prevent your health problems	0.922
	26. You know how to handle such health problems after this visit	0.916
General satisfaction	27. You were satisfied with this visit in general	0.925
	28. You would choose this hospital again if the need arises	0.883

**Table 4 ijerph-19-06916-t004:** Correlation matrix with AVE.

Dimension	Mean	SD	AVE	1	2	3	4	5	6
1. Environment/convenience	3.93	0.65	0.543	1					
2. Doctor-patient communication	4.13	0.60	0.679	0.807	1				
3. Health information	4.14	0.62	0.701	0.772	0.895	1			
4. Medical service fees	3.96	0.73	0.719	0.748	0.742	0.823	1		
5. Short-term outcome	4.10	0.67	0.845	0.707	0.787	0.818	0.787	1	
6. General satisfaction	4.18	0.65	0.819	0.736	0.798	0.805	0.758	0.863	1

**Table 5 ijerph-19-06916-t005:** Regression weights of parameters by multiple indicators, multiple causes model with structural equation modeling.

Link	Coefficient	95% Confidence Interval	*p* Value
Environment/convenience←mHealth app use	0.11	[0.05, 0.17]	<0.001
Health information←mHealth app use	0.06	[0.00, 0.11]	0.039
Medical service fees←mHealth app use	0.08	[0.03, 0.13]	0.004
Short-term outcome←mHealth app use	0.05	[0.00, 0.11]	0.054
General satisfaction←mHealth app use	0.02	[−0.03,0.08]	0.429
Doctor–patient communication←mHealth app use	0.05	[0.00, 0.10]	0.069
Environment/convenience←Age	−0.07	[−0.13, −0.01]	0.015
Environment/convenience←Education level	0.11	[0.02, 0.20]	0.021
Environment/convenience←Household income	0.03	[0.00, 0.07]	0.041
Environment/convenience←Area of residence	0.14	[0.02, 0.27]	0.023
Environment/convenience←First visit	−0.03	[−0.15, 0.08]	0.596
Health information←Age	−0.05	[−0.10, 0.01]	0.103
Health information←Education level	0.13	[0.04, 0.22]	0.006
Health information←Household income	0.05	[0.02, 0.09]	0.002
Health information←Area of residence	0.12	[0.01, 0.24]	0.035
Health information←First visit	−0.04	[−0.15, 0.07]	0.506
Medical service fees←Age	−0.10	[−0.16, −0.04]	<0.001
Medical service fees←Education level	0.15	[0.06, 0.24]	0.002
Medical service fees←Household income	0.05	[0.02, 0.08]	0.004
Medical service fees←Area of residence	0.15	[0.03, 0.27]	0.018
Medical service fees←First visit	0.03	[−0.08, 0.14]	0.587
Short-term outcome←Age	−0.06	[−0.12, 0.00]	0.056
Short-term outcome←Education level	0.10	[0.01, 0.18]	0.031
Short-term outcome←Household income	0.05	[0.01, 0.08]	0.005
Short-term outcome←Area of residence	0.08	[−0.03, 0.20]	0.164
Short-term outcome←First visit	−0.06	[−0.17, 0.04]	0.242
General satisfaction←Age	−0.06	[−0.12, 0.00]	0.038
General satisfaction←Education level	0.05	[−0.04, 0.14]	0.241
General satisfaction←Household income	0.06	[0.03, 0.10]	<0.001
General satisfaction←Area of residence	0.08	[−0.04, 0.19]	0.219
General satisfaction←First visit	−0.12	[−0.24, −0.01]	0.034
Doctor–patient communication←Age	−0.04	[−0.09, 0.02]	0.187
Doctor–patient communication←Education level	0.06	[−0.02, 0.15]	0.140
Doctor–patient communication←Household income	0.07	[0.04, 0.10]	<0.001
Doctor–patient communication←Area of residence	0.07	[−0.04, 0.19]	0.225
Doctor–patient communication←First visit	−0.06	[−0.17, 0.05]	0.265
mHealth app use←Age	−0.40	[−0.46, −0.34]	<0.001
mHealth app use←Education level	0.35	[0.26, 0.44]	<0.001
mHealth app use←Household income	0.08	[0.04, 0.11]	<0.001
mHealth app use←Area of residence	0.04	[−0.08, 0.16]	0.525
mHealth app use←First visit	−0.43	[−0.55, −0.31]	<0.001

## Data Availability

The datasets used and/or analyzed during the current study are available from the corresponding author on reasonable request.

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
