# Peer review of "The Association between mHealth App Use and Healthcare Satisfaction among Clients at Outpatient Clinics: A Cross-Sectional Study in Inner Mongolia, China"

_ijerph, 2022, doi:10.3390/ijerph19116916_

Round 1

Reviewer 1 Report

The authors have improved the new article over the previous article version. In addition, the study comprehensively discussed the satisfaction of users from the mHealth application. However, the information that makes this article stand out is still lacking.

1) Contributions of the study to previous research studies should be listed as items in the introduction.

2) Conclusion section is the section where the article is evaluated without prejudice by the authors. But the authors paid little attention to this section.

Author Response

Dear reviewer,

Thank you for your comments and suggestions.

The authors have improved the new article over the previous article version. In addition, the study comprehensively discussed the satisfaction of users from the mHealth application. However, the information that makes this article stand out is still lacking.

1) Contributions of the study to previous research studies should be listed as items in the introduction.

Response 1): We revised the last paragraph: “Previous studies have mainly reported the effects of mHealth app use on appointment scheduling, registration process via one-step regressions. Little is known about the comprehensive effects of the full range of hospital-based mHealth app use on healthcare satisfaction at outpatient department, such as doctor–patient communication, health information, medical service fees, short-term outcome. Structural equation modeling (SEM) is a powerful analytic tool to examine complex causal models among multiple variables simultaneously, and using latent factors to reduce measurement error, which is superior to one-step regressions. Our multidimensional analysis aims to examine the relationships between the full range of mHealth use and each domain of outpatient department satisfaction using SEM.”

2) Conclusion section is the section where the article is evaluated without prejudice by the authors. But the authors paid little attention to this section.

Response 2): We have improved the conclusion section “Data showed that clients of the study hospitals were satisfied with the services, but their satisfaction was not much associated with mHealth use. On the one hand, the healthcare system of hospitals should continue to keep such high satisfaction level. On the other hand, the limitation of the mHealth system should be improved to enhance communication and engagement among clients, doctors, and healthcare givers, as well as to pay more attention to health outcomes and satisfaction of clients.”

Reviewer 2 Report

This resubmitted paper is well improved. This study is to explore the association between mHealth app use and healthcare satisfaction among clients at outpatient clinics. Relevant data were collected through a cross-sectional study in Inner Mongolia, China. Overall study processes are sound and valid. Well reflected previous comments.

One thing to suggest is to enrich a conclusion, not just one sentence but at least one long paragraph.

Author Response

Dear reviewer,

Thank you for your comments and suggestions.

This resubmitted paper is well improved. This study is to explore the association between mHealth app use and healthcare satisfaction among clients at outpatient clinics. Relevant data were collected through a cross-sectional study in Inner Mongolia, China. Overall study processes are sound and valid. Well reflected previous comments.

One thing to suggest is to enrich a conclusion, not just one sentence but at least one long paragraph.

Response: We have improved the conclusion section: “Data showed that clients of the study hospitals were satisfied with the services, but their satisfaction was not much associated with mHealth use. On the one hand, the healthcare system of hospitals should continue to keep such high satisfaction level. On the other hand, the limitation of the mHealth system should be improved to enhance communication and engagement among clients, doctors, and healthcare givers, as well as to pay more attention to health outcomes and satisfaction of clients.”

This manuscript is a resubmission of an earlier submission. The following is a list of the peer review reports and author responses from that submission.

Round 1

Reviewer 1 Report

It is a multidimensional study that deals with the satisfaction of mHealth application in patients who apply to outpatient clinics. Some major corrections are needed.

1) There should be an image that summarizes the overall practice of the study.

2) Insights must be provided to enable patient-doctor communication.

3) The Conclusion section should summarize the work in general and include the conclusion.

4) The aims and findings of previous similar studies should be shared in the Introduction. A table containing comparisons with similar studies should be prepared.

5) At the end of the Introduction, there should be contributions that make this study different from previous studies.

Author Response

Response to Reviewer 1 Comments

It is a multidimensional study that deals with the satisfaction of mHealth application in patients who apply to outpatient clinics. Some major corrections are needed.

1) There should be an image that summarizes the overall practice of the study.

Response 1): We revised the introduction section and dicussion setion, the order was from mHealth use to satisfaction, and considered patient-doctor communication as key joint among them. mHealth services in hospital apps almost cover the outpatient treatment process, which is in line with all the aspects of satisfaction. SEM could be used to estimate this framework appropriately.

2) Insights must be provided to enable patient-doctor communication.

Response 2): We inserted two statements in the introduction section “Effective communication between doctor and patient is crucial for patient satisfaction[13]” and “The use of mHealth could facilitate doctor-patient communication and health outcomes[16, 17].” In the discussion section, we also added ”Maybe because the use rate of consultation and rating in mHealth was still low at 12% and 16%, respectively.” And in the limitation part  “especially for doctor-patient communication” was inserted. Finally, we revised a sentence in the conclusions “but also include dimensions to enhance the effective patient-doctor communication and health outcomes among clients”.

3) The Conclusion section should summarize the work in general and include the conclusion.

Response 3): We revised the conclusion section” The usage of services in hospital-based mHealth apps were imbalanced, majority focused on convenience of healthcare at outpatient clinics, which had different and limited effect on various domains of client satisfaction. mHealth industries should not only to make health care more convenient, but also include dimensions to enhance the effective patient-doctor communication and health outcomes among clients.”

4) The aims and findings of previous similar studies should be shared in the Introduction. A table containing comparisons with similar studies should be prepared.

Response 4): In the introduction section, we added “the previous studies mainly analyze mHealth use in terms of appointment with doctors, and to examine the general relationship among mHealth use and satisfaction by regressions[18, 20, 27].”  In the forth paragraph of dicussion setion, we have compared our results with those from previous studies.

5) At the end of the Introduction, there should be contributions that make this study different from previous studies.

Response 5): At the end of the Introduction, we added ” Our multidimensional analysis would like to find all joints between the whole spectrum of mHealth use and each domain of outpatient department satisfaction.”

Reviewer 2 Report

This study is to explore the relatinship between mHealth app usa and healthcare satisfaction in a country. The purpose is clear and the ovaerall study processes are proper. Followings are some comments to improve the quality of the manuscript.

  • In line 19, add p-value rather than (95% CI). 
  • In lines 20-21, add coefficients and p-values. 
  • Include some sentences of expected study implication at the end of the introduction
  • In line 82, complete a sentence of (1 representing the worst.  
  • In lines 87-88, more complete information should be provided. Most covariates are not 0-1 cases. 
  • In line 101, no need to provide a-pripri sanple size for SEM formular is necessary. Delete it.  
  • Provide correlations matrix with AVE.
  • In Table 1, it says n(%)/Mean (SD) but values provided in the table is not clear. For example, Age group 30 or below shows 504 (26.6). 504 is n 26.6 is %. There are not Mean and SD. Revise either item tile or values. 
  • Enhance conclusions including study limitations, study contribution  and study future directions.   

Author Response

Response to Reviewer 2 Comments

Comments and Suggestions for Authors

This study is to explore the relatinship between mHealth app usa and healthcare satisfaction in a country. The purpose is clear and the ovaerall study processes are proper. Followings are some comments to improve the quality of the manuscript.

1) In line 19, add p-value rather than (95% CI). 

Response 1): We revised as follows: “the coefficients β (p-value) on environment /convenience, health information, and medical service fees were 0.11 (p-value<0.001), 0.06 (p-value = 0.039 ) and 0.08 (p-value = 0.004)”

2) In lines 20-21, add coefficients and p-values. 

Response 2): We revised “However, app use was not significantly associated with satisfaction of doctor-patient communication (β=0.05, p-value=0.069), short-term outcomes (β =0.05, p-value=0.054), and general satisfaction (β=0.02, p-value=0.429).”

3) Include some sentences of expected study implication at the end of the introduction

Response 3): At the end of the introduction,we added “Our multidimensional analysis would like to find all joints between the whole spectrum of mHealth use and each domain of outpatient department satisfaction.”

4) In line 82, complete a sentence of (1 representing the worst.

Response 4): We completed ” (1 representing the worst, 5 representing the best). “

5) In lines 87-88, more complete information should be provided. Most covariates are not 0-1 cases. 

Response 5): We completed the statement “Covariates included social-demographic variables and first visit or not, i.e., age was divided into four groups (30 or below = 1, 31-45 = 2, 46-60 = 3, more than 60 = 4), gender (male= 1, female = 0) and area of residence (urban=1, rural=0) were dichotomous variables, education levels were coded as three categories (primary or no =1, middle school =2, college or higher =3), household monthly income was measured as the following groups: 0-2,000 RMB =1, 2,001-4,000 RMB = 2, 4,001-6,000 RMB = 3, 6,001-8,000 RMB = 4, 8,001-9,999 RMB =5, 10,000 RMB or more = 6, and an indicator variable defined as 1 for first-time visitors to the hospital, otherwise 0.”

6) In line 101, no need to provide a-pripri sanple size for SEM formular is necessary. Delete it.  

Response 6): We have deleted it.

7) Provide correlations matrix with AVE.

Response 7): We have revised a correlations matrix with AVE as shown in table 4.

8) In Table 1, it says n(%)/Mean (SD) but values provided in the table is not clear. For example, Age group 30 or below shows 504 (26.6). 504 is n 26.6 is %. There are not Mean and SD. Revise either item tile or values. 

Response 8): We have revised table 1. The variables needing mean and SD are moved to table 4.

9) Enhance conclusions including study limitations, study contribution  and study future directions. 

 Response 9): We revised the limitation seciton and conclusion seciton as follows:

  • This study was limited as a cross-sectional survey, which was conducted in a break between Covid-19 waves in China. However, we did not calculate the effect of it. Despite the diversity of demographic and socioeconomic characteristics in our sample, our findings were only from a stage of the ongoing mHealth development, further studies to follow up mHealth use, especially for doctor-patient communication are needed.
  • The usage of services in hospital-based mHealth apps were imbalanced, majority focused on convenience of healthcare at outpatient clinics, which had different and limited effect on various domains of client satisfaction. mHealth industries should not only to make health care more convenient, but also include dimensions to enhance the effective patient-doctor communication and health outcomes among clients.

Round 2

Reviewer 1 Report

I don't think the revisions made are enough. Therefore, the work must be completely reorganized.